# Research Advances in the Mutual Mechanisms Regulating Response of Plant Roots to Phosphate Deficiency and Aluminum Toxicity

**DOI:** 10.3390/ijms23031137

**Published:** 2022-01-20

**Authors:** Weiwei Chen, Li Tang, Jiayi Wang, Huihui Zhu, Jianfeng Jin, Jianli Yang, Wei Fan

**Affiliations:** 1College of Life and Environmental Sciences, Hangzhou Normal University, Hangzhou 311121, China; Chenweiwei@hznu.edu.cn; 2State Key Laboratory of Plant Physiology and Biochemistry, College of Life Sciences, Zhejiang University, Hangzhou 310058, China; wangjiayi0704@163.com (J.W.); 11907025@zju.edu.cn (H.Z.); lxm0404@zju.edu.cn (J.J.); 3College of Resources and Environment, Yunan Agricultural University, Kunming 650201, China; tangli7650@163.com; 4College of Horticulture and Landscape, Yunan Agricultural University, Kunming 650201, China

**Keywords:** acidic soil, aluminum toxicity, organic acids, phosphate deficiency, signal crosstalk, stress response

## Abstract

Low phosphate (Pi) availability and high aluminum (Al) toxicity constitute two major plant mineral nutritional stressors that limit plant productivity on acidic soils. Advances toward the identification of genes and signaling networks that are involved in both stresses in model plants such as *Arabidopsis thaliana* and rice (*Oryza sativa*), and in other plants as well have revealed that some factors such as organic acids (OAs), cell wall properties, phytohormones, and iron (Fe) homeostasis are interconnected with each other. Moreover, OAs are involved in recruiting of many plant-growth-promoting bacteria that are able to secrete both OAs and phosphatases to increase Pi availability and decrease Al toxicity. In this review paper, we summarize these mutual mechanisms by which plants deal with both Al toxicity and P starvation, with emphasis on OA secretion regulation, plant-growth-promoting bacteria, transcription factors, transporters, hormones, and cell wall-related kinases in the context of root development and root system architecture remodeling that plays a determinant role in improving P use efficiency and Al resistance on acidic soils.

## 1. Introduction

Occupying over 30% of the world’s arable lands, acid soils encompass a lot of factors constraining plant performance. Among others, mineral nutrition stress frequently occurs in acid soils because their solubility depends largely on soil pH [1]. In particular, phosphorus (P) deficiency and aluminum (Al) toxicity are usually interrelated, constituting two major plant nutritional stress factors that limit crop production on acidic soils [2,3,4]. Phosphorus is the structural component of phospholipids, nucleic acids, sugar phosphates, nucleotides, coenzyems, phytic acid, etc., and has a key role in reactions that involve ATP. Therefore, it is an essential micronutrient for plants and has many fundamental roles in plant growth and development. Although the total P content in most soils is high, the availability of inorganic P (Pi) that is the major form that plant roots absorb is often very low due to the low mobility and high fixation, particularly on acid soils in which high concentrations of cations such as Fe^3+^, Al^3+^, and Mn^2+^ fix P easily. On the other hand, when the soil pH is lower than 5.5, insoluble mineral Al can be released into soil solution in the ionic forms, mainly cationic Al^3+^, which is highly toxic to the plant root apex and consequently restrains the ability of roots to absorb water and nutrients [5,6,7,8,9]. Therefore, low P availability and Al toxicity greatly limit plant growth on acid soils that are distributed worldwide.

To meet the demand of food, an improper application of a large amount of fertilizers (especially nitrogen fertilizers) and the increase of acid deposition facilitates soil acidification [10]. Although the application of P fertilizer can improve the yield and quality of crops on acidic soils by improving P availability and reducing Al^3+^ activity, the efficiency of P fertilizer is economically low (the availability is only 10–25% in the current season). Moreover, excessive application of P fertilizer results in not only the exhaustion of P mineral resources but a series of environmental problems such as water eutrophication [11,12]. Therefore, to develop outstanding crop varieties through both molecular breeding and modern biotechnology methods is a smart solution to the problem of both P limitation and Al toxicity of crops on acid soils in the future.

Due to the coexistence of Al toxicity and P deficiency on acidic soils, researchers have been motivated to study the interaction between Al toxicity and P deficiency in plant adaptations to acidic soils. There are many excellent review papers that describe plant strategies to cope with either Al toxicity [13,14] or P deficiency [15], and the interactive effects of Al and P on plant adaptation to acid soils [3,4,16,17]. Recent evidence further showed that P efficiency is closely related to Al tolerance in ryegrass (*Lolium perenne* L.) [18]. On the other hand, root exudates stimulate colonization and chemotaxis of microorganisms in the rhizosphere where a lot of beneficial bacteria that are known as plant-growth-promoting rhizobacteria (PGPR) inhabit [19]. Many researchers have demonstrated that the use of PGPR could improve P nutrition by plants through secreting phosphatases and organic acids (OAs). The role of PGPR in alleviating Al toxicity has also been reported [20,21]. Interestingly, recent studies have shown that Al-tolerant phosphobacteria that were isolated from ryegrass have the ability to cope with both Al toxicity and P deficiency [18,22]. Moreover, by identifying chemicals that can overcome Al-induced meristem loss of Arabidopsis plants, a drug that is structurally related to a casein kinase2 (CK2) inhibitor, was found to be able to maintain the root stem cell niche under Al toxicity, indicating that CK2 activity is responsible for Al-induced root growth inhibition. Interestingly, CK2 activity inhibition prevented meristem loss that was induced by Pi deficiency, providing evidence that Al toxicity and P deficiency are both related to the cell cycle checkpoint [23].

Based on these findings, it is inferred that plants may have evolved some mutual mechanisms to cope with both Al toxicity and P deficiency stresses. Over the past two decades, considerable successful results have been achieved on plant P uptake, transport, distribution, and signal regulation under P deficiency, especially in model plants such as Arabidopsis (*Arabidopsis thaliana*), rice (*Oryza sativa*), and white lupin (*Lupinus albus*) [24,25,26]. Similarly, a series of Al-resistant genes that were identified from rice, wheat (*Triticum aestivum*), sorghum (*Sorghum bicolcor*), and Arabidopsis have opened up the understanding of the molecular mechanism of Al-resistance in plants [27,28,29].

With increasing research, accumulating evidence shows that Al-resistant genes may have versatile impacts in the context of the multiple stressors that are coexisting on acidic soils, including enhancing P acquisition efficiency, which is manifested in the fact that many functional genes may have co-evolutionary mechanisms in their regulatory pathways.

In this review paper, the advances in the common mechanisms including the secretion of OAs, plant-growth-promoting bacteria, cell wall properties, phytohormones, and iron (Fe) homeostasis which control Al resistance and P nutrition on acid soils, are summarized, providing insight into the potential solutions to maintain better P status and crop productivity on acid soils by changing root development and root architecture.

## 2. Physiological Mechanism of Plants Adapting to Al Toxicity and P Deficiency on Acidic Soils

The inhibition of root elongation has been regarded as the primary observable symptom of Al toxicity in plants, and, therefore, being widely used as a parameter for the evaluating Al toxicity and/or tolerance [30]. Plants have evolved both responsive and adaptive mechanisms to resist Al toxicity. These mechanisms could be grouped into two categories. One is the internal tolerance mechanism including complexation of Al by intracellular OAs, proteins, phenols, and other organic compounds, and sequestration into the vacuole. The other is the external exclusion mechanism including chelation of Al in rhizosphere by OAs and phosphate, adsorption and fixation of Al on cell wall, and induction of pH barrier in rhizosphere [17,31,32]. Compared to Al toxicity, the strategy of P uptake by plants is to enlarge the contact area between the roots and the soil so as to improve P acquisition efficiency. A great challenge for plants to enlarge the contact area is to change the root system architecture (RSA) that requires the following processes: the ratio of root to shoot is increased, the primary root growth is inhibited, and the lateral roots and root hairs are increased [33]. Moreover, plant roots secrete OAs such as malate and citrate under P deficiency stress, which can increase P solubilization. In addition, acid phosphatase that are secreted by roots plays an important role in hydrolyzing organic P in the rhizosphere [33,34]. Therefore, root OAs secretion and root development are conserved in response to both Al toxicity and P deficiency.

## 3. Convergent Evolution of Organic Acids for Plants Adapting to Al Toxicity and P Deficiency

Ionic Al triggers plant roots to specifically exude OAs to form Al-OA complexes in the rhizosphere, thus rendering Al non-toxic and making the roots 5–20 times more resistant to Al stress (Figure 1) [35]. In addition, OAs that are secreted by roots can form complexes with Al^3+^, Fe^3+^, and Ca^2+^ stronger than P [36,37], which increases plant P availability by releasing P from bound forms in soils (Figure 1). In some plant species, the more OAs that are secreted from the roots, the higher Al tolerance and P efficiency that have been achieved [35,38,39], suggesting that the release of OAs from roots is an effective strategy that is employed by plants for adapting to acidic soils.

The roles for OAs are not only limited to the external detoxification of Al and improvement of P availability in the rhizosphere but are also involved in the internal sequestration of Al^3+^ and release of free P within the cells. Cytosolic OAs can chelate Al^3+^ into non-toxic compounds, and, therefore, protect roots from Al^3+^ toxicity and, in turn, release more free P from Al-P precipitation for metabolic use (Figure 1). Studies on absorption kinetics showed that Al^3+^ could rapidly enter the root cells [40]. Transporters that are specific to Al^3+^ have evolved in plants so as to detoxify the remaining Al^3+^ in the apoplast by sequestering into vacuoles. For instance, in rice, active absorption of Al^3+^ is mediated by Nrat1, a member of the natural resistance-associated macrophage protein (Nramp) transporter family (Figure 1) [41]. Once entered into the cytoplasm, Al^3+^ can be further isolated into the vacuoles by a tonoplast-localized half-size ABC transporter, ALS1, possibly in the form of Al-OA complexes (Figure 1) [42]. In sorghum, SbNrat1, a close homolog of rice OsNrat1, also was shown to selectively transport Al^3+^ [43]. Further structure-function relationship analysis of two reported Nramp-type Al transporters in rice and sorghum revealed that members of the Nramp family with Al transport activity must have two specific motifs, Motif A (DPSN) and Motif B (AIIT) (Figure 2) [44]. Surprisingly, based on motif analysis, no Nramp-type Al transporter exists in *Brachypodium distachyon* (Figure 2), which, in the evolution of the Pooideae diverged just prior to the majority of important temperate cereals and forage grasses. Furthermore, no Nramp member has these two characteristic motifs in Arabidopsis and tomato (Figure 2), suggesting that there is no Nramp-type Al transporter in dicots. In Arabidopsis, the aquaporin member NIP1;2 is responsible for the absorption of Al-malate from the root cell wall into the root symplasm, with subsequent Al xylem loading and root–shoot translocation (Figure 1) [45]. When expressed in yeast, NIP1;2 has bidirectional permeability to the Al-malate complex [45,46]. Whether functional homologs of NIP1;2 are present in other dicots has to be investigated.

Additionally, since a large amount of P is stored in the vacuoles, P deficiency drives vacuolar P transport into cytosols to meet metabolic demands [47,48]. This process in rice has been demonstrated to be mediated by two inorganic P-transporters OsVPE1 (vacuolar Pi efflux transporter1) and OsVPE2 located in tonoplast (Figure 1) [49]. Since vacuolar OAs act as ligands to replace P in bonds with excessive metal ions such as Ca^2+^ and Fe^3+^ [50], thereby facilitating free P release in the vacuole and subsequent transport into cytosols (Figure 1). This might be one reason why Al toxicity and P deficiency can independently stimulate the increase of OA concentrations in plant species that are well adapted to acidic soils [51,52,53].

## 4. Molecular Mechanism of Plants Adapting to P Deficiency and Al Toxicity on Acidic Soils

### 4.1. Al-Activated Malate and Citrate Transporters

The first bona fide Al tolerance gene is wheat TaALMT1 that was isolated from a pair of isogenic wheat lines ET8 and ES8 [54]. TaALMT1 encodes an aluminum-activated malate transporter that belongs to a previously unknown protein family. It is noteworthy that TaALMT1 is functionally active in the absence of extracellular Al^3+^, although its transport ability could be further enhanced by Al. Therefore, members of ALMT family do not necessary require Al^3+^ to play their roles in mediating malate or other anions flux across the different kinds of membranes [55]. Later, Arabidopsis AtALMT1, which encodes Al-activated malate transporter, was identified [56], and many homologous genes have been characterized from a variety of plants species [28]. Both TaALMT1 and AtALMT1 are plasma membrane-localized anion channel proteins that mediate malate secretion from the root tips [56,57,58]. Although the molecular determinants that are involved in the binding of Al^3+^ to the ALMT protein remain unknown, structural modification and phylogenetic studies on ALMTs indicate that several different domains in both TaALMT1 and AtALMT1 proteins are likely to act together in the Al-mediated enhancement of transport activity [54,59,60]. Recently, the cryo-electron microscopy structure of AtALMT1 has been analyzed and provides the structural basis for Al-activated malate transport by AtALMT1 [61]. The AtALMT1 forms a homodimer to assemble an anion channel, each subunit contains six transmembrane helices (TMs) and six cytosolic α-helices. There are two pairs of Arg residues that are located in the center of the channel pore and contribute to malate recognition. Al binding to the extracellular of AtALMT1 results in the conformational changes of the TM1-2 loop and the TM5-6 loop, resulting in the extracellular gate [61].

In addition to transport malate, a second transport substrate and regulatory mechanism of TaALMT1 has been discovered [62,63]. It was found that TaALMT1 also has a high permeability to gamma-aminobutyric acid (GABA), a non-protein amino acid that is involved in plant signaling cascades. Interestingly GABA is not only transported by TaALMT1, but also regulates the activity of transporters. Similarly, the apoplastic pH and anion composition also seem to regulate TaALMT1 transport activity, which can be stimulated by increasing the anion concentration and/or more alkaline apoplastic conditions [62]. These functional characteristics provide additional regulatory mechanisms for the Al-mediated regulation of TaALMT1 activity. It will be interesting to know the structural basis for Al-activated malate transport and GABA transport by TaALMT1 in the future.

The transporters that are responsible for Al-activated citrate secretion belong to a protein family that is different from the ALMT family. The first Al-activated citrate transporter gene was identified by map-based cloning in sorghum [64] and barley [65]. Phylogenetic analysis showed that both genes belong to an already known transporter protein family, multidrug and toxic compound extrusion (MATE), therefore, named SbMATE1 for sorghum and HvMATE1 (also named HvAACT1) for barley. The homologs of SbMATE and HvAACT1 were functionally characterized in Arabidopsis (AtMATE1) [66], maize (ZmMATE1) [67], wheat (TaMATE) [68,69], rice bean (VuMATE1 and VuMATE2) [70,71], and rice (OsFRD2 and OsFRDL4) [72,73], and indicates that this subgroup of MATE transporters mediate citrate transport. It is worth noting that the citrate transporter had been characterized before the characterization of SbMATE and HvMATE1. Arabidopsis FRD3 mutants showed constitutive Fe deficiency responses no matter if Fe was sufficient or deficient in the growth medium [74]. Map-based cloning of FRD3 demonstrated that this gene belongs to the MATE family, and both electrophysiology and transgenic approaches have demonstrated that FRD3 is a citrate transporter that facilitates Fe translocation from the roots to the shoots [75]. While FRD3 is a pericycle-localized citrate transporter that mediates citrate flux into the xylem, AtMATE1 is localized mainly at epidermis and cortex, thereby facilitating citrate efflux into the rhizosphere to chelate Al^3+^ [66]. A similar scenario has also been reported in barley where a 1-kb insertion in the upstream of the HvAACT1 coding region occurred in the Al-tolerant barley accessions, thereby enhancing its expression and altering the location of expression to the root tips to detoxify Al^3+^ [76].

Not only transport substrates, but also transport characteristics differ between ALMTs and MATEs. When several MATE transporters that were involved in Al resistance were expressed in *Xenopus oocyte* system, these proteins mainly mediated the constitutive pH-dependent citrate transport and were not activated by extracellular Al^3+^ [64,67,69,70,77]. Further electrophysiological analysis showed that these MATE transporters mediate an electrogenic transport resulting from the influx of a large number of cations (H^+^, Na^+^, and/or K^+^) in the absence of exogenous intracellular citrate. Recently, Doshi et al. (2017) demonstrated that sorghum SbMATE has a wide range of substrate recognition in *Xenopus oocyte* and yeast cell systems, mediating the efflux of ^14^C-citrate anion that is driven by H^+^ and/or Na^+^ and organic monovalent cation, ethidium [77]. It is also interesting to know the structural basis of Al-activated citrate secretion for MATE proteins in the future.

Although P deficiency-induced OAs secretion has long been recognized [37], the association of OAs secretion to the expression of ALMT and MATE has not been well evaluated. GmALMT1 and TaALMT1, were speculated to be involved in plant P efficiency. In the soybean P-efficient genotype HN89, GmALMT1 was shown to be up-regulated along with the increased release of malate from roots in response to P deficiency [78]. Overexpressing wheat TaALMT1 in barley promoted P acquisition of transgenic plants grown on acidic soils [79]. The improved P acquisition was ascribed mainly to the role of TaALMT1 in maintaining root growth under soil acidity, which results likely from Al resistance [79]. However, the observed greater P uptake per unit root length in TaALMT1-expressing barley lines could be also due to the fact that malate is released into the rhizosphere, thus favoring P mobilization from acidic soils and subsequent uptake [79]. When acidic soil was neutralized by lime, the saturation of Al decreased greatly and the grain yield of the transgenic and non-transgenic lines were similar, suggesting that enhanced P uptake under soil acidity was indeed largely achieved as an indirect effect of TaALMT1 enhancing Al resistance. These observations suggest that plant P availability and Al resistance may co-evolve through environmental selection of ALMT proteins.

### 4.2. Regulatory Factors Involved in ALMT1 and MATE1 Expression

The first transcription factor (TF) that was identified that regulates the expression of ALMT and MATE was Arabidopsis AtSTOP1 (Sensitive to Proton Rhizotoxicity1). This TF was cloned by mutant screening of hypersensitivity to low pH. AtSTOP1 is a Cys2His2 Zinc-finger TF and regulates the expression of AtALMT1, AtMATE1, and ALUMINUM SENSITIVE 3 (ALS3) [66,80]. In addition, AtSTOP1 also regulates the expression of GLUTAMATE DEHYDROGENASE-1 and -2 (GDH1 and GDH2) [81] and NRT1.1 [82] to regulate low pH tolerance. As low pH is closely related to both Al toxicity and P availability, it is possible that the regulation of genes that are involved in low pH contributes to both Al tolerance and P nutrition. In rice, OsART1, an AtSTOP1 ortholog, seems to regulate a set of genes that are different from that in Arabidopsis, including OsNrat1 (Al^3+^ transporter), OsMGT1 (Mg^2+^ transporter), and OsFRDL2 (citrate transporter) [41,71,83], but not regulate genes that are related to low pH. This difference might be due to the large difference in low pH tolerance between rice and Arabidopsis. As an ammonium preferred plant, rice is very tolerant to low pH, whereas Arabidopsis is a sensitive one [84].

Recent studies have found that AtSTOP1/AtALMT1 pathway is not limited to its role in Al and H^+^ resistance, but in a P starvation response [85,86]. An antagonism has previously been established between phosphate and Fe availability [87]. Under P deficiency, LOW PHOSPHATE ROOT 1 (LPR1), a ferroxidase, promotes Fe deposition and callose production in the meristem and the elongation zone, which results in the Fe^3+^-dependent ROS production, and consequently, the inhibition of root growth [85,86]. It was found that P deficiency-induced AtALMT1-mediated malate secretion is critical for apoplastic Fe accumulation and ALMT1-mediated malate secretion in the rhizosphere relies on STOP1. As the inhibition of primary root growth is accompanied by the enhancement of lateral root proliferation under low P conditions [88,89], ALMT1-mediated malate secretion can eventually increase the P uptake on acidic soils by remodeling of the RSA. However, a recent study demonstrated that light, or more precisely blue light, is required for malate-mediated Fe accumulation in the root apoplast [90]. Therefore, previous studies using Petri dish-grown Arabidopsis seedlings should be revisited to determine their relevance to plants that are grown in nature [91]. Interestingly, a more recent study showed that blue light signal perception at the shoot and transduction to the root via cryptochromes and their downstream signaling factors is involved in P deficiency-induced root growth inhibition [92].

Although it is well known that AtSTOP1 is a core regulator regulating the expression of both AtALMT1 and AtMATE1, our understanding on how Al stress or P deficiency regulates AtSTOP1 is limited [93]. Recently, by screening factors affecting AtALMT1 expression, several new regulators have been identified. First, an F-box proteins REGULATION OF ATALMT1 EXPRESSION 1 (RAE1) and RAE1 HOMOLOG 1 (RAH1) were identified to participate in STOP1 ubiquitination and promote its degradation via the ubiquitin-26S proteasome pathway, which is important for balancing Al resistance and plant growth [94,95]. Second, SUMOlyation and deSUMOlyation are involved in balancing the activity of STOP1, in which SUMO E3 ligase SIZ1 and SUMO proteases EARLY IN SHORT DAYS 4 (ESD4) antagonistically regulates the ability of STOP1 to promote AtALMT1 expression [96,97]. Third, THO has been found to be involved in regulating the stability and activity of STOP1. THO is a multisubunit complex that is present in yeast (*Saccharomyces cerevisiae*), plants, and animals, which functions in transcription, mRNA processing, and export [98,99,100]. The plant THO complex is composed of at least six subunits: HPR1/THO1/EMU1, THO2, TEX1/THO3, THO5A/B, THO6, and THO7A/B. Recently, Guo et al. (2020) found that HPR1 is involved in the regulation of Al resistance and low Pi response partly through the modulation of nucleocytoplasmic STOP1 mRNA export [101]. Another core component of the THO complex, RAE2/TEX1, has also been found to regulate Al resistance and low Pi response [102]. However, RAE2 did not affect STOP1 mRNA accumulation in the nucleus, although STOP1 protein level was reduced in rae2. These results suggest that response to both Al toxicity and low P availability are conservatively evolved through the regulation of STOP1 protein stability, which further demonstrate the core position of STOP1 in both stresses responses.

Besides STOP1, other TFs have been identified to be involved in the expression regulation of ALMT1. For instance, Ding et al. (2013) reported that AtWRKY46 is able to repress AtALMT1 expression, while Al induced expression repression of AtWRKY46 itself [103]. Tokizawa et al. (2015) identified CALMODULIN BINDING TRANSCRIPTION ACTIVATOR2 (CAMTA2) to be involved in the expression regulation of AtALMT1 [104]. Recently, Zhu et al. (2021) reported that CALMODULIN-LIKE 24 (CML24) interacts with CAMTA2 to promote AtALMT1 expression, and each could interact with WRKY46, thereby suppressing the transcriptional repression of AtALMT1 by WRKY46 [105]. However, in this study, the expression of AtWRKY46 was found to be induced by Al stress [105]. Considering that AtALMT1 expression was completely abolished in the stop1 mutant, it seems that other TFs play, at most, minor roles in regulating the expression of AtALMT1 or represents the secondary effects on STOP1/ALMT1 modules. Nevertheless, it is interesting to know whether these TFs are also involved in the P deficiency response by regulating AtALMT1 expression.

### 4.3. Two ABC Transporters Are Involved in Response to P Deficiency and Al Toxicity

Another example has been observed regarding Al resistance genes to be involved in the P deficiency response. In this case, two ABC transporter proteins STAR1/ALS3 (STAR2) were first reported to be implicated in Al resistance. Larsen et al. (1996) screened a recessive Al-sensitive mutant als3 that showed hypersensitivity to Al stress compared to wild type plants [106]. Map-based cloning identified ALS3 as an ABC transporter that is located in the root cortex, leaf sheath, and phloem [107,108]. Usually, ABC transporters contain both a nucleotide (ATP)-binding domain and a transmembrane (TM) domain [109], whereas ALS3 lacks an ATP binding domain which is necessary for ABC transporters to function. This finding led the researchers to suspect that there is another ABC transporter interacting with ALS3 to operate, and an ABC transporter AtSTAR1 that contains only an ATP domain but no TM domain that was involved in Al resistance was characterized in Arabidopsis [110]. AtSTAR1/ALS3 complex possibly mediated Al efflux from the outer cell layers of the root tip [110]. Similarly, rice OsSTAR1, a homolog of AtSTAR1, forms an ABC complex with OsSTAR2 (which contains the TM domain) to regulate Al resistance. The STAR1/2 protein complex is localized in vesicle membranes and shows efflux transport activity that is specific for UDP-glucose when expressed in *Xenopus laevis* oocytes. However, the exact mechanism underlying the OsSTAR1/OsSTAR2 complex-mediated Al resistance is not clear. UDP-Glc is an activated form of glucose that is used as a substrate for glycosyltransferases to synthesize various glycosides [111], the authors speculated that STAR1-STAR2 that are located on the membrane may be responsible for transporting UDP-Glc from the cytosol into the vesicles. UDP-Glc or glycoside that is derived from UDP-Glc would then be released from the vesicles to the apoplast by exocytosis and used to modify the cell walls to mask the sites for Al binding, resulting in Al tolerance in rice [112]. Recently, the homologs FeSTAR1 and FeSTAR2 in buckwheat were identified, and the complex of FeSTAR1/FeSTAR2 seemed to be located in the vesicles, which may be involved in cell wall polysaccharide metabolism through UDP-glucose transport, thus affecting Al tolerance in buckwheat [113,114]. However, whether they play a role in response to P deficiency in buckwheat remains to be studied.

Interestingly, the STAR1/ALS3 module was also involved in the P deficiency response in Arabidopsis, in which Fe homeostasis is involved. Under low P conditions, lpr1 mutants exhibited decreased Fe accumulation in roots, resulting in the insensitivity of root growth to low P [86,87,115]. Interestingly, both atstar1 and als3 mutants also exhibited insensitivity of root growth to P deficiency [115]. However, the AtSTAR1/ALS3 pathway involves UDP glucose, which can reverse the excessive accumulation of Fe^3+^ and rescue the short root phenotype in the als3 mutant when suffering from P deficiency [115]. Moreover, ALS3 acts upstream of STOP1/ALMT1 and LPR1 in controlling primary root growth under Pi deficiency by repressing STOP1 protein accumulation in the nucleus [116]. The accumulation of STOP1 in the nucleus of the als3 mutant is dependent on Fe and Al^3+^ under P deficiency, indicating that Fe^2/3+^ and Al^3+^ act similarly to increase the stability of STOP1 [116]. Considering that AtSTOP1 is the major regulator of AtALMT1 and ALS3 expression, it appears that the effects of als3 mutation on STOP1 is a feedback of Pi deficiency response. Further studies showed that Fe inhibits the degradation of STOP1 via RAE1-mediated 26S proteasome under P deficiency [94]. These results suggest that there may be antagonism between Al resistance that is conferred by ALS3/STAR1 and P acquisition. Overall, the root development changes that are induced by ALMT1/STOP1 and AtSTAR1/ALS3 seem to be a low P-specific response. The physiological mechanism centered on Fe homeostasis may be mediated by ALMT1/STOP1 and AtSTAR1/ALS3, which mediate different Al resistance pathways to regulate root remodeling under low P conditions. If these responses are also applicable to other crops on acidic soils, it is easy to speculate that the presence of Al toxicity centered on Fe and Al oxides and the soil chemical link between Al toxicity and low P availability, which is centered on the presence of Fe and Al oxides, may lead to co-selective pressure for pleiotropic mechanisms enabling plants to be able to both tolerate Al^3+^ and to acquire P more efficiently.

### 4.4. Hormone-Mediated Interaction between P Deficiency and Al Stress

It has been well-documented that phytohormones such as auxin, ethylene, cytokinin, and jasmonic acid (JA) play an important role in regulating root apical meristem (RAM) activity. Since both Al toxicity and P deficiency affect RAM activity, signaling crosstalk of different hormones on Al-P interactions has attracted great attention. There are a lot of studies on the role of phytohormones on either Al toxicity or Pi deficiency response, but this review focuses solely on studies relating phytohormones to both stresses or on those providing clues relating phytohormones with both stresses.

Both Al toxicity and P availability affect the synthesis and transport of auxin, leading to changes in auxin distribution and root growth [117,118,119]. Moreover, the insensitivity of the primary root inhibition in the lpr mutant in response to P deficiency is achieved through auxin regulation [120,121,122]. Interestingly, *lpr1-1 lpr2-1* double mutants also showed significantly longer primary roots compared with WT under Al toxic conditions [123]. These results suggest that the primary root growth inhibition that is caused by P deficiency and Al toxicity could be attributed to a common auxin regulatory pathway [122]. As auxin is an important regulator of the cell cycle of root meristem, it is also possible that auxin signaling is involved in the CK2-mediated stem cell activity regulation. In addition, the auxin receptor transport inhibitor response1 (TIR1) and auxin response factor ARF not only participate in the regulation of lateral root formation under low P condition [124] but affects the lateral root density and primary root growth under Al stress [123], indicating that there is also a common auxin signaling pathway in P deficiency and Al stress-mediated lateral root development (Figure 3).

Ethylene has been shown to mediate the response to Pi deficiency and regulate the inhibition of primary root growth, promotion of lateral root elongation, and root hair formation [125,126]. Al stress also activates ethylene signaling in the primary roots as evidenced by ethylene signaling mutants etr1-3 and ein2-1 that exhibit reduced Al-induced root inhibition (Figure 3) [119]. Al stress also resulted in the accumulation of ethylene during sunflower seed germination [127]. In Arabidopsis, the local synthesis of auxin in the root transition zone (TZ) induced by Al stress depends on ethylene signal and is mediated by two key auxin synthesis enzymes family genes YUCCA (YUC) and TAA, which are involved in tryptophan-dependent auxin synthesis, eventually leading to the growth inhibition of the primary root (Figure 3) [128,129]. These studies suggest that ethylene-dependent auxin biosynthesis plays a key role in Al-induced inhibition of root growth. However, Al stress-induced auxin synthesis is considered to be related to the elongation of the primary root in maize [117]. Therefore, more experimental evidence is needed to explain whether this contradiction is due to differences in species or experimental conditions. In addition, of the 12 ACS genes that are involved in ethylene biosynthesis, ACS2 and ACS6 were highly up-regulated by P deficiency and Al toxicity. At the same time, these two stresses also enhanced the expression of ACO1 and ACO2, suggesting that ACS and ACO may be one of the causes of ethylene bursts that are triggered by P deficiency and Al toxicity in Arabidopsis (Figure 3) [119,130].

It has been well recognized that the production of IAA is widespread among soil bacteria [131]. In addition to OAs, root exudates contain high amounts of amino acids such as Trp, which can be taken up by bacteria in the rhizosphere to synthesize indole-3-acetic acid (IAA), some of which is, in turn, taken up by the plant. The addition of L-tryptophan to growth media improved the P-solubilizing activity of phosphate solubilizing bacteria that were able to produce IAA [132]. This effect was related to the dorp of pH and exudation of OAs [132]. On the other hand, some ACC, an ethylene biosynthesis precursor, is released by the roots and taken up by bacteria, resulting in the expression induction of ACC deaminase. Experimental evidence showed that the amount of ACC deaminase that is produced by phosphate solubilizing bacteria was significantly associated with the liberation of Pi from Ca-P when ACC was the sole N source [133]. Therefore, it appears that PGPR represents a central hub for regulating auxin and ethylene balance to cope with both Al toxicity and P deficiency.

Al stress induces local accumulation of auxin in primary roots and further triggers excessive accumulation of cytokinins, which ultimately results in the inhibition of primary root growth [134]. Among them, cytokinin accumulation is a dependent process of adenosine phosphate isoprene transferase (IPT). The up-regulation of IPT that is induced by Al and the cytokinin response were significantly enhanced in the yuc1D mutant, while this response was significantly reduced in the arf7/19 double mutant, suggesting that this is an auxin dependent process (Figure 3) [134].

JA has also been reported to be associated with P deficiency [135]. In the JA signaling mutant coi1-34, low P-induced root inhibition was partly relieved. Therefore, JA signaling that is mediated by COI1 at low P level can lead to growth inhibition of the main roots (Figure 3) [135]. In comparison, the JA signal is regulated by ethylene under Al stress, which mediates the inhibition of Al-induced root growth by regulating microtubule polymerization and malate secretion that is mediated by ALMT1 (Figure 3) [136].

The discovery of melatonin could date back to 1995, in which two groups simultaneously identified the presence of melatonin (*N*-acetyl-5-methoxytrytamine) in vascular plants for the first time [137,138]. Plant melatonin is involved in growth, rooting, seed germination, photosynthesis, osmoregulation, and protection against abiotic and biotic stressors [139]. Importantly, the recent identification of CAND2/PMTR1, a phytomelatonin receptor in Arabidopsis (Arabidopsis thaliana) provided critical evidence for melatonin to be regarded as a plant hormone [140]. A recent study reported that melatonin could alleviate Al toxicity by reducing ROS and enhancing the exclusion of Al from the root apex by altering cell wall polysaccharides in wheat [141]. However, direct evidence lacks with respect to the role of melatonin in low P deficiency response. It has been shown that the transcript levels of some YUCCA and TAA genes could be significantly repressed by melatonin treatment [142,143]. What’s more, genes encoding auxin-influx carriers (AUX1/LAX), which regulate lateral root development, root gravitropism, and root hair development [144], were also down-regulated by melatonin [145,146]. Sun et al. (2021) proposed that auxin signaling and melatonin signaling are indirectly connected and possibly converging at nitric oxide [147]. These results suggest that melatonin could be an important regulator mediating both Al toxicity and P deficiency response. However, the direct evidence has to be provided (Figure 3).

### 4.5. Wall-Associated Kinases Are Involved in the Regulation of P Uptake and Al Tolerance

There is tremendous experimental evidence that has demonstrated that cell wall polysaccharides, especially pectin, play critical roles in both Al toxicity and P nutrition [148,149,150,151,152,153]. It is easy to envision that cell wall pectin containing a lot of carboxyl groups have high binding capability to Al^3+^, thereby regulating Al resistance. At first glance, it, however, seems counterintuitive that phosphate anions could have interactions with cell wall polysaccharides. One possibility could be that cell wall polysaccharides, which has been shown to be a cation absorber, has a particularly high affinity for cations [148], and thus can facilitate the remobilization of P that is deposited in the cell wall [154]. However, the molecular basis on how cell wall pectin metabolism relates to both Al toxicity and P deficiency is much less understood. Recently, several lines of evidence suggest that wall-associated kinases (WAKs) could be involved in cell wall pectin-mediated responses to both Al toxicity and P deficiency. Wall-associated kinases (WAK) are a subfamily of receptor kinases (RLKs)/Pelle superfamily, which also includes other subfamilies such as WAK kinases (WAKL) and leaf rust 10 disease-resistance locus receptor-like protein kinase (LRK10) [155,156,157]. WAKs family proteins usually contain a cysteine-rich (Cys-rich) galacturonan binding domain (GUB_Wak), epidermal growth factor (EGF) repeats, a TM domain, as well as a cytoplasmic serine/threonine kinase domain, which can span the plasma membrane and extend out into the cell wall [158,159,160,161]. WAKs have been shown to play a role in cell development, morphogenesis, and resistance to environmental stimuli [162,163,164,165,166].

Sivaguru et al. (2003) found that the expression of AtWAK1 in Arabidopsis could be rapidly induced by Al, and overexpressing AtWAK1 could improve Al resistance of transgenic plants [162]. However, how AtWAK1 could relate to Al resistance was not known. Recently, Lou et al. (2020) found that a rice bean NAC-type TF, VuNAR1, is able to bind to the promoter of AtWAK1 when ectopically expressed in Arabidopsis, thereby promoting AtWAK1 transcription [167]. The expression of AtWAK1 is negatively associated with cell wall polysaccharides content that is critical for binding of Al [166], providing evidence that links a WAK protein to pectin metabolism. A knockout of the glycine-rich protein, AtGRP3, which interacts with AtWAK1 has also been shown to enhance Al resistance in Arabidopsis [166,168]. However, the expression of AtGRP3 was not induced by Al, and the grp3 mutant had a long root phenotype without Al. Therefore, whether Al-induced grp3 root growth inhibition is due to the mechanism enhancing Al resistance or represents a secondary effect of grp3 mutations remains to be validated. However, whether AtWAK1 plays roles in P deficiency responses also remains unknown.

A second piece of evidence comes from Hufnagel et al. (2014) [169] who found that Sorghum SbPSTOL1 that is homologous to rice serine/threonine receptor kinase OsPSTOL1 [170] participated in the increase of root surface area, thus increasing P acquisition and grain yield under low P availability soils. Interestingly, SbPSTOL1 shared similar GUB_Wak and TM domains, intron-exon structures, and genomic localization with AtWAK1 [169]. Recent studies have shown that amino acids in the GUB_Wak domain can covalently bind with natural pectin and galacturonic oligosaccharides in the cell wall [160,161,164,171,172]. Therefore, SbPSTOL1 protein may function as WAKs, as a receptor for the activation of the signaling cascade in response to extracellular stimuli such as P deficiency. Considering the role of SbPSTOL1 in promoting sorghum root growth and P acquisition, these proteins can jointly control Al resistance and P availability. Nevertheless, the direct evidence on the role of PSTOL1 in Al tolerance has to be investigated.

## 5. Conclusions and Perspectives

Healthy root development and ‘ideal’ RSA play a determinant role for plants to better resist Al toxicity and P deficiency stress on acidic soils. OAs are the key substances for plants to resist both Al toxicity and P deficiency on acidic soils. They not only chelate Al to render it non-toxic in rhizosphere and in the cytoplasm, but also solubilizes P in rhizosphere and releases P that is stored in the vacuoles to increase P utilization efficiency. Recent advances on pleiotropic effects of STOP1/ALMT1 and STAR1/ALS3 have provided us with some substantial information at the molecular level and confirmed that there are indeed some synergistic regulatory centers and checkpoints (such as root development) between the two stresses. However, their connection is more based on the understanding of Al-tolerant genes to extend to P deficiency. Whether the known P signaling pathways, such as PHR, SPX, PHO, and PFH, are also involved in Al-tolerant regulation (especially in the regulation of OAs secretion) needs further study.

One of the common responses to P deficiency and Al toxicity is the inhibition of primary root growth and promotion of lateral root growth. Plant hormones, especially auxin and ethylene, have been found to play an important role in controlling root responses to P deficiency and Al toxicity. In the context of the remodeling of primary root elongation and lateral root emergence, it seems likely that the common signaling pathways exist for the two stresses (Figure 1 and Figure 3). However, the expression patterns of auxin transporter genes vary among different species, and hormones in promoting or inhibiting root growth have thresholds differences which results in changes in root development in response to P deficiency and Al toxicity among different species. Therefore, it is of particular importance to further study how hormones improve Al tolerance and P utilization efficiency by regulating the expression of genes. For example, rice Al-resistant transcription factor ART1 regulates at least 30 genes that are related to the internal and external detoxification of Al [173]. However, it is not clear whether hormone-mediated signaling pathways interplay with the Al signal transduction pathway that is regulated by ART1. Although experimental evidence has demonstrated the impact of Al resistant genes on crop performance on acidic soils, the effects of these genes on P utilization efficiency in field crop varieties remains to be studied. Moreover, most of the knowledge about the regulation of interaction between P deficiency and Al toxicity comes from Arabidopsis. Therefore, it is also urgent to fill in the gap between the basic research and the creation of new varieties in breeding projects. The identification of Al-resistant genes through genetic variation in crop varieties is the most suitable molecular tool for cultivating high-yield crops on acidic soils. With the deepening of knowledge and the advance in related technology, it is likely there will be a design and cultivation of “intelligent” crops with ideal roots, which can not only obtain P, but also resist Al toxicity on acidic soils, thus fundamentally reducing P fertilizer consumption and achieving sustainable agricultural development. Also, better understanding of PGPR regulation in response to both Al toxicity and P deficiency will be helpful for developing efficient bacteria-based biofertilizers, which provides an alternative solution to improve plant performance on acid soils.

## Figures and Tables

**Figure 1 ijms-23-01137-f001:**
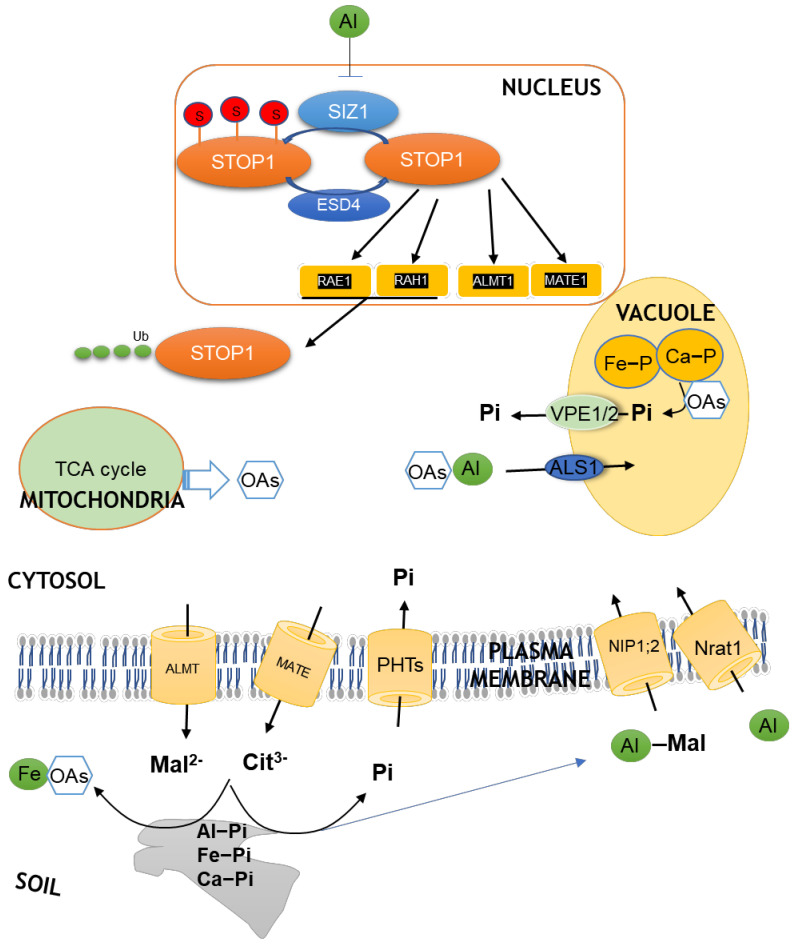
Schematic illustration of the roles of organic acid anions (OAs) and their metabolism regulation in Al detoxification and P solubilization on acidic soils. The OAs are produced mainly through mitochondria tricarboxylic acid (TCA) cycle. Under stress conditions, stress signals trigger the transcription factor STOP1 via SUMOylation and ubiquitination regulation to induce the transcription of transporter genes that are responsible for OAs exudation, Pi uptake, and Al complexation and uptake. The OAs that are secreted from roots function in against Al toxicity and P deficiency when grown on acid soils. OAs form complexes with Al^3+^ to protect the roots from Al toxicity in the rhizosphere, and solubilize rhizosphere Pi via complexation with Al, Fe, and Ca oxides and hydroxides on mineral surfaces. ALMT: Aluminum-activated malate transporter; ESD4: EARLY IN SHORT DAYS 4; MATE: Multidrug and toxic compound extrusion; NIP1;2: nodulin 26-like intrinsic protein 1;2; Nrat1: NRAMP-type Al transporter1; PHTs: Phosphate transporters; SIZ1: a SUMO E3 ligase; RAE1: REGULATION OF ATALMT1 EXPRESSION 1; RAH1: RAE1 HOMOLOG 1; VPE: vacuolar Pi efflux transporter.

**Figure 2 ijms-23-01137-f002:**
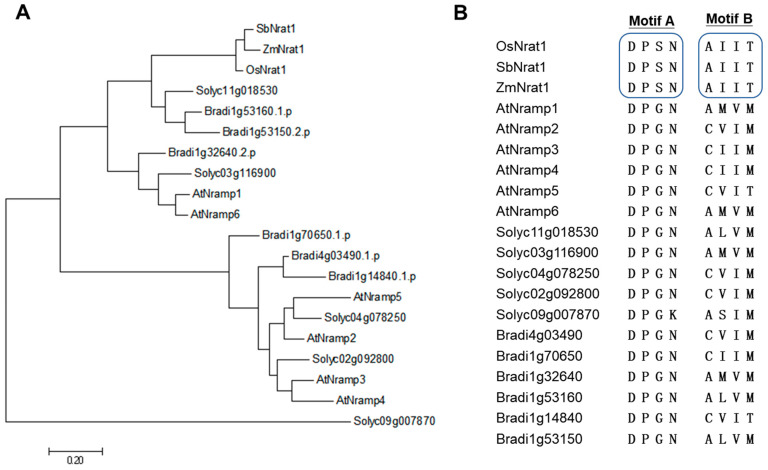
Phylogenetic and motif analysis for the 20 plant metal transporters of the Nramp family. (**A**) An unrooted phylogenetic tree was built by MEGA7. (**B**) Sequence alignment of the two signature motifs of the 20 plant Nramp transporters. The 20 Nramp transporters are rice OsNrat1 (XP_015625418), Sorghum SbNrat1 (XP_002451480), Maize ZmNrat1 (NP_001334019), six Arabidopsis Nramp proteins, five tomato Nramp proteins, and six Brachypodium distachyon Nramp proteins.

**Figure 3 ijms-23-01137-f003:**
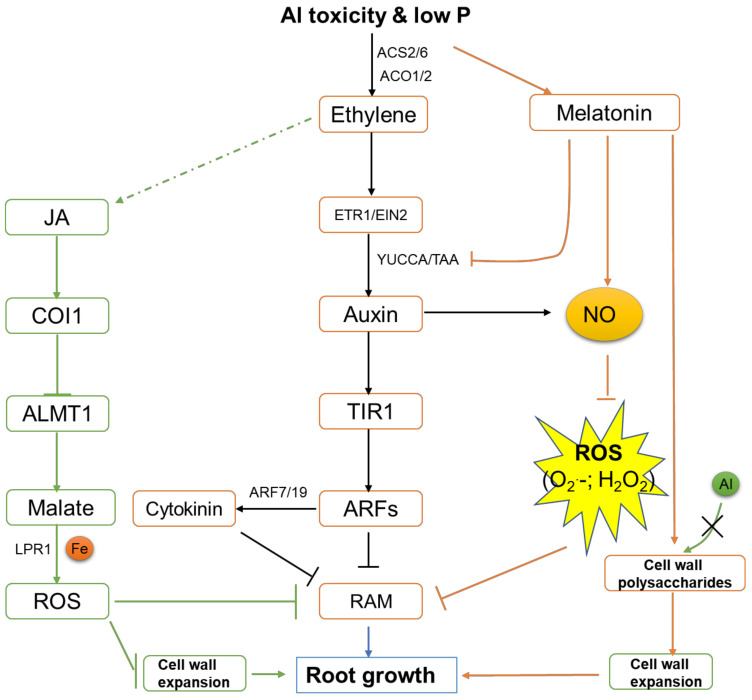
A simplified model for the relationship among different hormones in response to both Al toxicity and low P. Al stress and P deficiency induce the local up-regulation of TRYPTOPHAN AMINOTRANSFERASE OF ARABIDOPSIS 1 (TAA1) and YUCCA (YUC) in the root transition zone through an ethylene-dependent pathway, which contributes to auxin accumulation in the root transition zone (TZ), suppressing primary root growth. Local cytokinin (CK) accumulation acts downstream of auxin signaling through ARF7/9, synergistically regulating the stress-induced inhibition of root growth. COI1-mediated jasmonate (JA) signaling is involved in stress-induced root growth inhibition through ALMT1-mediated malate exudation from roots to affect ROS production. In addition, the production of melatonin under stress conditions is able to reduce the ROS production and induce cell wall expansion to promote root growth. JA: jasmonic acid; COI1: F-box protein CORONATINE INSENSITIVE 1; ALMT1: aluminum-activated malate transporter1; LPR1: low phosphate-resistant root1; TIR1: transport inhibitor response 1; ARF: auxin responsive factor; RAM: root apical meristem; NO: nitric oxide; ROS: reactive oxygen species; ETR1: ethylene receptor1; EIN2: ETHYLENE INSENSITIVE2.

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
