# Peer review of "Research Advances in the Mutual Mechanisms Regulating Response of Plant Roots to Phosphate Deficiency and Aluminum Toxicity"

_ijms, 2022, doi:10.3390/ijms23031137_

Round 1

Reviewer 1 Report

There are various typos and grammatical mistakes and the whole manuscript requires a thorough readout preferably from a native English speaker e.g. line 53 “biotechnology methods is an smart solution”

Lien 86 “2. Results Physiological mechanism of plants”

Reference citation and references require a thorough attention

Some latest papers are suggested for consultation and update

Lambers, H., 2022. Phosphorus acquisition and utilization in plants. Annual review of plant biology, 73.

Cai, S., Wu, L., Wang, G., Liu, J., Song, J., Xu, H., Luo, J., Shen, Y. and Shen, S., 2022. DA-6 improves sunflower seed vigor under Al3+ stress by regulating Al3+ balance and ethylene metabolic. Ecotoxicology and Environmental Safety, 229, p.113048.

Vance, W., Pradeep, K., Strachan, S.R., Diffey, S. and Bell, R.W., 2021. Novel sources of tolerance to aluminium toxicity in wild cicer (Cicer reticulatum and Cicer echinospermum) collections. Frontiers in plant science, 12.

Wei, Y., Han, R., Xie, Y., Jiang, C. and Yu, Y., 2021. Recent advances in understanding mechanisms of plant tolerance and response to aluminum toxicity. Sustainability, 13(4), p.1782.

Chandra, J. and Keshavkant, S., 2021. Mechanisms underlying the phytotoxicity and genotoxicity of aluminum and their alleviation strategies: A review. Chemosphere, p.130384.

Phukunkamkaew, S., Tisarum, R., Pipatsitee, P., Samphumphuang, T., Maksup, S. and Cha-Um, S., 2021. Morpho-physiological responses of indica rice (Oryza sativa sub. indica) to aluminum toxicity at seedling stage. Environmental Science and Pollution Research, pp.1-11.

Line 469 “5. Conclusions and Remarks” to “Conclusions and perspectives”

Author Response

There are various typos and grammatical mistakes and the whole manuscript requires a thorough readout preferably from a native English speaker e.g. line 53 “biotechnology methods is an smart solution”

Response: revised.

Lien 86 “2. Results Physiological mechanism of plants”

Response: revised.

Reference citation and references require a thorough attention

Some latest papers are suggested for consultation and update

Lambers, H., 2022. Phosphorus acquisition and utilization in plants. Annual review of plant biology, 73.

Cai, S., Wu, L., Wang, G., Liu, J., Song, J., Xu, H., Luo, J., Shen, Y. and Shen, S., 2022. DA-6 improves sunflower seed vigor under Al3+ stress by regulating Al3+ balance and ethylene metabolic. Ecotoxicology and Environmental Safety, 229, p.113048.

Vance, W., Pradeep, K., Strachan, S.R., Diffey, S. and Bell, R.W., 2021. Novel sources of tolerance to aluminium toxicity in wild cicer (Cicer reticulatum and Cicer echinospermum) collections. Frontiers in plant science, 12.

Wei, Y., Han, R., Xie, Y., Jiang, C. and Yu, Y., 2021. Recent advances in understanding mechanisms of plant tolerance and response to aluminum toxicity. Sustainability, 13(4), p.1782.

Chandra, J. and Keshavkant, S., 2021. Mechanisms underlying the phytotoxicity and genotoxicity of aluminum and their alleviation strategies: A review. Chemosphere, p.130384.

Phukunkamkaew, S., Tisarum, R., Pipatsitee, P., Samphumphuang, T., Maksup, S. and Cha-Um, S., 2021. Morpho-physiological responses of indica rice (Oryza sativa sub. indica) to aluminum toxicity at seedling stage. Environmental Science and Pollution Research, pp.1-11.

Response: thanks and we have included these references into the text.

Line 469 “5. Conclusions and Remarks” to “Conclusions and perspectives”

Response: Thanks for your suggestion.

Reviewer 2 Report

A fine review  with uptodate referencing  relating to an old problem. 

Appreciate the connections you make   of which there are many such as the triangle of Fe Al and P

maybe you could summarize where monocots and dicots differ and overlap

few spelling problems but this reads well in English  thanks 

you could provide a separate diagram highlighting new thoughts on cell wall associations here    recent advances show the wall to be much more than a static frame work for the plant   

there are a number of sticky notes with comments   one repeat theme is putative roles root associated microbes in these responses      their activities overlap with the mechanisms you describe  and an extension of a section into this area seems justified       it stimulated my thinking 

Author Response

A fine review with uptodate referencing relating to an old problem.

Appreciate the connections you make of which there are many such as the triangle of Fe Al and P

Response: Thanks for your positive assessment on our current review paper.

maybe you could summarize where monocots and dicots differ and overlap

Response: We don’t find there are obvious difference in mechanisms by which  monocots and dicots cope with Al toxicity and P deficiency. For example, ALMT1 and MATE1 proteins are present in both monocots and dicots. Whether a plant species secrete malate or citrate in response to Al toxicity or P deficiency is also not related to monocots or dicots. Thanks for your concern.

few spelling problems but this reads well in English thanks

Response: thanks for your positive evaluation on our writing, some spelling errors have been corrected.

you could provide a separate diagram highlighting new thoughts on cell wall associations here recent advances show the wall to be much more than a static frame work for the plant

Response: A very good point. Yes, we also thought to provide a diagram. However, its seems that recent evidence is insufficient to support a model to illustrate the importance of cell wall properties in Al toxicity and P deficiency.  

there are a number of sticky notes with comments one repeat theme is putative roles root associated microbes in these responses their activities overlap with the mechanisms you describe and an extension of a section into this area seems justified it stimulated my thinking

Response: Many thanks for your notes, we have made revisions as possible as we can. As to the microbes, we have added some papers related to microorganisms into the text. However, we are not familiar with this field, we feel difficult to made an extensive review on this area.

Reviewer 3 Report

The thesis in Introduction should be extended. The main idea of the paper should be a clear statement regarding to the aims of the review – what the authors intend to prove or illustrate?

The authors included many quite old (more than 10 years old) references but not introduce many recent papers – below only a few example of papers which were omitted in discussion:

https://pubmed.ncbi.nlm.nih.gov/30077920/

https://www.ncbi.nlm.nih.gov/pmc/articles/PMC5992465/

https://academic.oup.com/plcell/article-abstract/33/4/1361/6119328?redirectedFrom=fulltext

https://journals.plos.org/plosone/article?id=10.1371/journal.pone.0190900

https://www.sciencedirect.com/science/article/abs/pii/S1673852716301710

https://www.mdpi.com/1422-0067/19/10/3073/htm

In my opinion current form of review paper does not allow to evaluate literature, identify patterns and trends in the literature or identify research gaps and recommend a new research areas.

In this matter I recommend to review current literature again. I think that table form would be good for presentation of recent studies of P-deficiency/Al-toxicity interaction.

I think that the paper could be extended by adding paragraph about the endophytic and rhizosphere bacteria role in Al-toxicity and P-deficiency plant stress.

The figures should be well described in captions – lack explanation of abbreviations

Additionally there are many misprints in the text and unnecessary capital letters.

Author Response

The thesis in Introduction should be extended. The main idea of the paper should be a clear statement regarding to the aims of the review – what the authors intend to prove or illustrate?

Response: Thanks for your suggestions. We have made some extension to the Introduction. The main idea has been stated in the last paragraph of Introduction. In this review paper, the advances in the common mechanisms including secretion of OAs, plant growth promoting bacteria, cell wall properties, phytohormones, and iron (Fe) homeostasis, which controlls Al resistance and P nutrition on acid soils are summarized. providing insight into the potential solutions to maintain better P status and crop productivity on acid soils by changing root development and root architecture.

The authors included many quite old (more than 10 years old) references but not introduce many recent papers – below only a few example of papers which were omitted in discussion:

https://pubmed.ncbi.nlm.nih.gov/30077920/ included

https://www.ncbi.nlm.nih.gov/pmc/articles/PMC5992465/

https://academic.oup.com/plcell/article-abstract/33/4/1361/6119328?redirectedFrom=fulltext

https://journals.plos.org/plosone/article?id=10.1371/journal.pone.0190900

https://www.sciencedirect.com/science/article/abs/pii/S1673852716301710

https://www.mdpi.com/1422-0067/19/10/3073/htm

In my opinion current form of review paper does not allow to evaluate literature, identify patterns and trends in the literature or identify research gaps and recommend a new research areas.

In this matter I recommend to review current literature again. I think that table form would be good for presentation of recent studies of P-deficiency/Al-toxicity interaction.

Response: Many thanks for your suggestion. We have added some recently published papers including those your proposed. The research on either Al toxicity or P deficiency has been well documented and summarized in some previous review papers for example, Kochian et al. (2015; Annual review of Plant Biology 66, 571–598). This present review focuses on some new findings connecting both stresses, which has been summarized through Figures.

I think that the paper could be extended by adding paragraph about the endophytic and rhizosphere bacteria role in Al-toxicity and P-deficiency plant stress.

Response: Thanks for your constructive suggestions. We have added some papers related to microorganisms into the discussion. However, we are not familiar with this field, we feel difficult to made an extensive review on this area. Hope our revisions are acceptable.

The figures should be well described in captions – lack explanation of abbreviations

Response: figure captions have been added. Thanks.

Additionally there are many misprints in the text and unnecessary capital letters.

Response: the typos have been corrected. Thanks.

Reviewer 4 Report

The review is interesting and could merit publication, but the present version requires huge improvement. There are many typos and mistakes in the use of english. I.e:

21--> With emphasis on

47--> acidification.

52--> eutrophization.

76--> with increasing research

86--> Delete "results"

92--> sequestration instead of "complexation"

106--> convergent ecvolution

122--> Al toxicity; injury is not a common term in this context.

134--> motifs.

136--> poaceae

167--> require

and many others, so please correct all these typos and mistakes.

Another major concern is that authors describe a lot of genes, something that makes the reading complicated for a non-specialist reader, thus narrowing the scope of audience. Authors should include a table with a summary of all the genes mentiones and with a brief summary of their function and the reference. 

Author Response

The review is interesting and could merit publication, but the present version requires huge improvement. There are many typos and mistakes in the use of english. I.e:

21--> With emphasis on

47--> acidification.

52--> eutrophization.

76--> with increasing research

86--> Delete "results"

92--> sequestration instead of "complexation"

106--> convergent ecvolution

122--> Al toxicity; injury is not a common term in this context.

134--> motifs.

136--> poaceae (it’s Pooideae not poaceae in the original paper)

167--> require

and many others, so please correct all these typos and mistakes.

Response: thanks for your interesting in our review and for your carefully check on our writing problems. The errors your pointed out has been revised.

Another major concern is that authors describe a lot of genes, something that makes the reading complicated for a non-specialist reader, thus narrowing the scope of audience. Authors should include a table with a summary of all the genes mentiones and with a brief summary of their function and the reference. 

Response: thanks for your concern and suggestion. Although many genes are involved in this review paper, these genes have been well summarized previously (for example, Kochian et al., 2015 https://doi.org/10.1146/annurev-arplant-043014-114822; Zhang et al., 2019 Int. J. Mol. Sci. 2019, 20, 1551; doi:10.3390/ijms20071551). This review paper does not focus on the functional description of these genes but on new clues of these genes in alleviating both Al toxicity and P deficiency. So, we feel it not suitable for us here to organize a table for these genes once again.

Round 2

Reviewer 1 Report

The authors have incorporated almost all the comments. Just reference citation (41 citations) is not as per the format of the journal. Please see 56, 60, 71, ....

Author Response

Thank you. The paper has been carefully checked and revised

Reviewer 3 Report

The paper was improved and can be published in IJMS.

Author Response

Thank you again for your positive comments. Some small mistakes in the article have been carefully revised

Reviewer 4 Report

A major improve has been performed. I recommend publication

Author Response

(The authors gave the same response as above.)
